# Cooperative and Escaping Mechanisms between Circulating Tumor Cells and Blood Constituents

**DOI:** 10.3390/cells8111382

**Published:** 2019-11-03

**Authors:** Carmen Garrido-Navas, Diego de Miguel-Pérez, Jose Exposito-Hernandez, Clara Bayarri, Victor Amezcua, Alba Ortigosa, Javier Valdivia, Rosa Guerrero, Jose Luis Garcia Puche, Jose Antonio Lorente, Maria José Serrano

**Affiliations:** 1GENYO, Centre for Genomics and Oncological Research (Pfizer/University of Granada/Andalusian Regional Government), PTS Granada Av. de la Ilustración, 114, 18016 Granada, Spain; carmen.garrido@genyo.es (C.G.-N.); diego.miguel@genyo.es (D.d.M.-P.); ci.bayarri@gmail.com (C.B.); albaortigosa@correo.ugr.es (A.O.); jlpuche@ugr.es (J.L.G.P.); jlorente@ugr.es (J.A.L.); 2Laboratory of Genetic Identification, Department of Legal Medicine, University of Granada, Av. de la Investigación, 11, 18071 Granada, Spain; 3Integral Oncology Division, Virgen de las Nieves University Hospital, Av. Dr. Olóriz 16, 18012 Granada, Spain; jose.exposito.sspa@juntadeandalucia.es (J.E.-H.); victor.amezcua.md@gmail.com (V.A.); jvaldib@gmail.com (J.V.); mjs@ugr.es (R.G.); 4Department of Thoracic Surgery, Virgen de las Nieves University Hospital, Av. de las Fuerzas Armadas, 2, 18014 Granada, Spain

**Keywords:** circulating tumor cells, tumor cell dissemination, immune system, microbiome

## Abstract

Metastasis is the leading cause of cancer-related deaths and despite measurable progress in the field, underlying mechanisms are still not fully understood. Circulating tumor cells (CTCs) disseminate within the bloodstream, where most of them die due to the attack of the immune system. On the other hand, recent evidence shows active interactions between CTCs and platelets, myeloid cells, macrophages, neutrophils, and other hematopoietic cells that secrete immunosuppressive cytokines, which aid CTCs to evade the immune system and enable metastasis. Platelets, for instance, regulate inflammation, recruit neutrophils, and cause fibrin clots, which may protect CTCs from the attack of Natural Killer cells or macrophages and facilitate extravasation. Recently, a correlation between the commensal microbiota and the inflammatory/immune tone of the organism has been stablished. Thus, the microbiota may affect the development of cancer-promoting conditions. Furthermore, CTCs may suffer phenotypic changes, as those caused by the epithelial–mesenchymal transition, that also contribute to the immune escape and resistance to immunotherapy. In this review, we discuss the findings regarding the collaborative biological events among CTCs, immune cells, and microbiome associated to immune escape and metastatic progression.

## 1. Introduction

The presence of circulating tumor cells (CTCs) in the peripheral blood has been largely associated with reduced disease-free and overall survival [1,2,3]. Even though metastasis is a highly inefficient process (tumor cell survival is less than 0.01%) [4], it is responsible for the majority of cancer-associated deaths [5]. In fact, it is accepted that CTCs are the initiator factor of metastatic relapse and their presence identifies patients with a higher risk of developing metastasis [6,7]. However, the complex biological processes enabling CTCs to survive and disseminate is not yet well understood and little is known about the cellular and genetic events involved both in the metastatic initiation and in its progression.

The success of the metastatic process is conditioned by the established relationship between tumor cells and the surrounding microenvironment. During the metastatic process, tumor cells interact with the immune system, which modulates this process [8]. The immune system has a dual role, both repressing but also promoting cancer progression. In fact, formation of CTC clusters or microemboli, not only composed of CTCs but also leukocytes, cancer-associated fibroblasts, endothelial cells, and platelets, was shown to facilitate the metastatic process and thus be related to poorer outcome in patients with breast [9] and gastric cancer [10], among others.

In this review, we will focus our interest in the “intimate friendship" between CTCs and the immune system. This private alliance benefits tumor progression through CTCs survival in this hostile microenvironment, the blood.

However, we cannot forget that the microenvironment is not only composed by immune system cells, stromal cells, and components of the extracellular matrix (ECM). CTCs and microbes co-evolve inside the ecosystem within our bodies [11,12] as will be further described in Section 3. This interaction influences the activity of the immune system on cell survival and expansion of CTCs [13].

In this review, we evaluate the current literature on interactions among CTCs, immune system cells, and microbiome in the tumor progression. We discuss how immune cells–CTC interactions contribute to the survival of these CTCs and how the microbiome can promote this positive association, finally supporting the metastatic process.

## 2. Promotion of Circulating Tumor Cells through the Immune System

The immune system is educated to eliminate the foreign and to respect the innate [14]. However, in the case of cancer, tumor cells are able to use the immune system to facilitate their own survival and migration. This phenomenon is known as concomitant immunity (CI) [15,16].

The plasticity of the immune system is well known and thus, according with the tumor type, the functional contribution of each immune cell can also be different [17,18]. However, some immune events are intimately associated with promotion of cancer, independently on the tumor type. Inflammation is one of these events and it is recognized as one of the “hallmarks" of cancer [19]. This process involves different types of immune cells, among which platelets, macrophages, and neutrophils can be highlighted [20,21].

Platelets are anucleated blood cells with a diameter of 2–4 µm originated during megakaryocytes maturation in the bone marrow and circulate in large numbers (1.5–4.0 × 10^9^/L) in the bloodstream [22,23]. Platelets are the main cells involved in thrombosis and hemostasis, thus, related with the physiological and pathophysiological processes occurring during inflammation [24]. Interestingly, several studies have reported their role in cancer progression, especially during cancer metastasis [25] as they actively promote the metastatic process. Metastasis-promoting mechanisms affected by platelets are related to both migration of tumor cells and cancer cell survival in circulation [26].

Regarding the migration process, platelets store large amounts of transforming growth factor β (TGFβ), which is associated with an increase of the invasion potential of tumor cells. Thus, tumor cells-conjugated platelets release mediators to modify blood vessels permeability, including dense granule-release, histamine, eicosanoid metabolites, or serotonin [23]. These mediators induce endothelial cell retraction, exposing the basement membrane, and thus facilitate cancer cell extravasation [27]. In addition, platelets activation by cancer cells lead to the generation of platelet-derived microparticles (PDMPs), which can also release mediators like TxA2 and 12-HETE. These metabolites may enhance cell migration and invasion, eventually increasing the metastatic potential of cancer cells [28].

However, self-migration ability of CTCs is not enough to complete a successful metastatic process, survival of these cells, once in the blood, depends on the formation of circulating microemboli [9] as well as the acquisition of resistant phenotypes to the surrounding microenvironment. The acquisition of these phenotypes involves a biological process known as epithelial to mesenchymal transition (EMT) [29]. The EMT process explains how tumor cells change their phenotype, allowing them to detach, invade, and metastasize through the blood or lymphatic systems. Among others, the EMT involves loss of E-cadherin, disrupting cell-to-cell adhesions and altering gene expression by increasing β-catenin nuclear localization [30]. In contrast, N-cadherin, which is highly expressed in mesenchymal cells, fibroblasts, neural tissue, and cancer cells, is elevated during EMT. This cadherin switch, from E-cadherin to N-cadherin, is closely associated with the increased invasiveness, motility and metastasis potential of tumor cells. Moreover, activated platelets induce EMT through secretion of growth factors and cytokines (e.g., TNFα and TGFβ) [31]. Interestingly, these cytokines are also associated with the inflammatory process as previously explained [32].

Furthermore, CTCs-conjugated platelets also coordinate the engagement of other immune cells during the dissemination process [33]. In fact, CTCs-conjugated platelets recruit neutrophils, macrophages, and other immune cells through release of chemokines, such as CXCL5 or CXCL7. Among white blood cells (WBC), neutrophils are recognized as the mediators of metastasis initiation [34,35]. Neutrophils promote tumor development by initiating an angiogenic switch and facilitating colonization of CTCs. In fact, some groups support the idea that WBC shape a protective cover around CTCs, avoiding their recognition and destruction by other immune cells [36]. It has long been known that circulating platelet–neutrophil complexes are present in a wide range of inflammatory conditions including cancer. In this interaction, neutrophils are responsible to activate platelets and it was shown that the neutrophils–platelets interaction initiates inflammatory responses [37]. Platelets interact with neutrophils by multiple intermediates including platelet P-selectin binding to neutrophil P-selectin glycoprotein ligand-1 (PSGL-1) [38]. In addition, it has been suggested that the neutrophils-platelets complexes interacting to CTCs bring the latter to the endothelium, which is an essential step in hematogenous dissemination metastasis [34]. Thus, platelets prime tumor cells to promote neutrophil extracellular traps (NETs) formation, which are also involved in endothelial activation [39].

However, the interaction between the platelets–tumor cells complex and immune cells is not only restricted to neutrophils. The release of CXCL12, which is highly present in platelets, allows recruitment of CXCR4-positive cells such as macrophages to prepare the metastatic niche for CTCs [40]. Neutrophils are the first leukocytes to be recruited in response to chemotactic signaling and are responsible for stimulating the repair process and initiating inflammation. This influx is followed by monocytes, which, upon entry into the tissue, differentiate into macrophages. These macrophages promote invasion and metastasis from the primary tumor site through their ability to engage cancer cells in an autocrine loop that promotes cancer cell [41]. This autocrine signaling triggers cancer cells to produce CSF-1, which promote epidermal growth factor (EGF) production by macrophages. Finally, cancer cells and macrophages co-migrate towards tumor blood vessels, where macrophage-derived VEGF-A promotes cancer cell intravasation [42]. In addition, tumor migration is upregulated by macrophage-derived cathepsins, SPARC, or CCL18, that enhance tumor cell adhesion to extracellular matrix proteins [41]. Finally, CTCs produce CCL2 that recruits inflammatory monocytes, which in turn increase vascular permeability and allow migration of these tumor cells [16].

Nevertheless, migration and survival of CTCs belong together, so the promotion of CTC migration alone is not enough to allow metastasis. Anoikis is a programmed cell death induced by cell detachment [43] and essential for CTC survival. Another effect of the collaboration between immune cells and CTCs includes the protection of CTCs from anoikis [44]. Platelets are involved in this protective mechanism as it was observed that they induce RhoA-(myosin phosphatase targeting subunit 1) and MYPT1-protein phosphatase (PP1)-mediated Yes-associated protein 1 (YAP1) dephosphorylation and nuclear translocation, resulting in apoptosis resistance [45]. Apoptosis signal-regulating kinase 1 (Ask1) is a stress-responsive Ser/Thr mitogen-activated protein kinase kinase kinase (MAP3K) in the Jun N-terminal kinases (JNK) and p38 pathways. Once Ask1 levels are reduced in platelets, active phosphorylation of protein kinase B (Akt), JNK and p38 is downregulated, and thus tumor metastasis is attenuated [46].

In conclusion, the fate of CTCs is not to survive alone but with help of their mates within the immune system and thus, survival of CTCs depends on their ability to interact with immune cells. However, this favorable interaction between CTC and immune cells depends also on the status of our gut microbiota that is intimately linked with the nature of the immune system.

## 3. Survival of Circulating Tumor Cells Through the Interaction of Microbiota with the Immune System

The evolution of any disease, including cancer, depends highly on the physiological status of the host. The gut microbiota has emerged as an important factor of health and disease [47]. Likewise, our microbiota conditions the status of our immune system [48]. In fact, gut microorganisms are involved in the immune system development and in the response of the host against different pathologies, like cancer. Taken together, tumor cells-microbiome-immune system interactions may improve the likelihood of cell survival and induce tumor cell migration (Figure 1) [49].

As it has been mentioned before, the inflammatory process is an essential step in the development and progression of cancer. Microbes have a critical role in the initiation and maintenance of chronic inflammatory conditions [50,51]. However, how do microbes influence on the inflammation process and on the migration and survival of CTCs?

The gut microbiota contributes to cancer progression through different mechanisms. Recently, it was demonstrated that both, DNA-damaging superoxide radicals or genotoxins produced by the gut bacteria could initiate colon cancer. In addition, bacteria may induce cell proliferation through interactions with T-helper cells or Toll-like receptors, respectively [52]. In colon cancer patients, an increase of the *Escherichia coli* population was observed to induce colitis and colibactin synthesis and thereby, to promote inflammation.

Furthermore, it has been demonstrated that microbes as *Bacillus* sp., *Enterococcus faecium,* and *E. coli* produce peptides which alter host epithelial growth factor, activating intracellular pathways associated to migration. In a pioneer work, Wynendaele, E et al. [53] discovered that certain quorum sensing peptides produced by bacteria (molecules that microbes use to coordinate their gene expression and behavior) interact with cancer cells. This study demonstrated that Phr0662 (*Bacillus* sp.), EntF-metabolite (*E. faecium*), and EDF-derived (*E. coli*) peptides can initiate HCT-8/E11 colon cancer cell invasion. According to results of this group, the Phr0662 peptide targets epidermal growth factor receptors (EGFR and ErbB2). Upregulation of EGFR induces activation of the Ras/raf/MEK/MAPK, PI3K/Akt, and STAT intracellular signaling cascades [54] altering gene transcription and allowing migration of tumor cells. However, despite this work being extremely interesting, it is only a preliminary and exploratory in vitro assay and more exhaustive analyses including cancer patients should be carried out to validate these results.

Microbes can also alter cancer cell epigenetics through production of metabolites affecting gene expression [55]. This is the case of *Bifidobacterium* spp., which produces folate, one of the most powerful methyl donors involved in gene silencing. Thus, the gut microbiome is also involved in chromatin remodeling via acetylation and deacetylation of histones through butyrate production. Butyrate is a common metabolite of the microbiome, inducing cell differentiation via histone acetylation of the intestinal T reg cells [56].

Moreover, the microbiome plays an important role in the epithelial mesenchymal transition (EMT), an essential step for CTC migration and survival. In fact, microbes produce toxins that contribute to EMT [57]. Some of those microbes, as *Bacteroides fragilis*, *Fusobacterium nucleatum*, and *E. faecalis*, clear E-cadherin from epithelial cells, a transmembrane adhesion protein, leading to colonic epithelial proliferation [58]. Most of the studies on the interaction of the microbiome with cancer cells have been developed on colon cancer, murine models, or in vitro assays. Likewise, in a recent study including a murine model, colonic epithelial cells were transformed to express Ly6A/E, a stem cell marker implicating mesenchymal features, and Doublecortin-like kinase 1 (DCKL1), a marker of cancer, by the presence of *E. faecalis* [59]. DCLK1 is a member of the protein kinase super family and the doublecortin family, which is overexpressed in many cancers, including colon, pancreas, liver, esophageal, and kidney cancers. It is now suggested to be a master regulator of pluripotency factors, including Nanog, Oct4, Sox2, Klf4, and Myc, that are critical for stemness of cancer cells (CSC, cancer stem cells) and EMT transcriptional factors, including Snail, Slug, Twist, and Zeb 1 [60]. Interestingly, all these markers are involved in regulation of both EMT and CSCs and are controlled by DCLK1 expression in cancer models [61]. Furthermore, Westphalen, CB et al. [62], reported that DCLK1 induces quiescence of tumor cells. Quiescence is a common property of CSCs that is associated with the EMT process as a critical step for the migration and progression of tumor cells [63]. In consequence, EMT allows not only CTCs migration and tumor relapse, but also, induces the ability of CTCs to escape the immune system cancer treatments (Figure 1) [64].

Another biological mechanism used by the microbiome to enhance cancer progression includes the modulation of the immune system. Among all the microbes involved in this process, the enterotoxin *Bacteriodes fragilis* (ETBF) stands up due to the activation of STATA3 and T helper cells, both with an important role in the inflammatory process [65]. In fact, Chung, L et al. [65], demonstrated that *Bacteroides fragilis* toxin (BFT) can activate a pro-carcinogenic inflammatory cascade, related to IL-17R, NF-κB, and Stat3 signaling, in colonic epithelial cells (CECs). Likewise, the activation of NF-κB in these cells, induces other chemokines as CXCL1 that mediates recruitment of CXCR2-expressing polymorphonuclear immature myeloid cells, promoting ETBF-mediated distal colon tumorigenesis. Another bacterium associated with poor oncological outcomes is *Fuscobacterium nucleatum*. It has been suggested that *F. nucleatum* promotes tumorigenesis through both pro-inflammatory and immunosuppressive effects. Furthermore, *F. nucleatum* is associated with activation of cytokines IL-6, IL-12, IL-17, and TNF-α, which cooperatively upregulate NFκB, a critical regulator of cellular proliferation [66]. Some studies associated the presence of high levels of *F. nucleatum* with the EMT process [67]. In this context, Mima, K et al. [68], identified that *F. nucleatum* adheres to and invades epithelial cells mainly through the virulence factors, including Fusobacterium adhesin A, Fusobacterium autotransporter protein 2, and fusobacterial outer membrane protein. To the contrary, other studies raise their skepticism about the role of *F. nucleatum* on EMT, as it still remains unclear whether *F. nucleatum* triggers the colonic EMT process. Ma, CT, et al., showed that *F. nucleatum* infection did not affect expression levels of E-cadherin and β-catenin [69]. However, it was associated with proliferation and invasion of colon cancer cells as it significantly increased phosphorylation of p65 (a subunit of nuclear factor-κB), as well as expression of interleukin (IL)-6, IL-1β and matrix metalloproteinase (MMP)-13. Regardless of the fact that there are not any explicit studies evaluating the direct action of the microbiota on CTCs, we here reviewed some of the biological processes in which microbes alter tumorigenic pathways. As they are involved in inflammation or inducing EMT, both biological processes intimately associated with the ability of CTCs to migrate and to survive, we suggest the potential interaction between them.

## 4. Conclusions and Perspectives

In conclusion, the complex interactions between the microbiome, the immune system and CTCs may allow us to grasp the insights of the dissemination process occurring in cancer and the immune system´s mechanisms involved in this process. Therefore, the interactions among microbiome, immune system, and CTCs could aid the rational design of interventions that strengthen the antimetastatic potential of combined treatments to prevent appearance of metastasis. Moreover, emerging evidences may provide new mechanisms to control the dissemination process through the development of new therapeutic strategies with the microbiota as target. However, this topic is still an incipient area of research and further investigation is needed to clarify the association of the microbiome with the immune system and the dissemination process.

## Figures and Tables

**Figure 1 cells-08-01382-f001:**
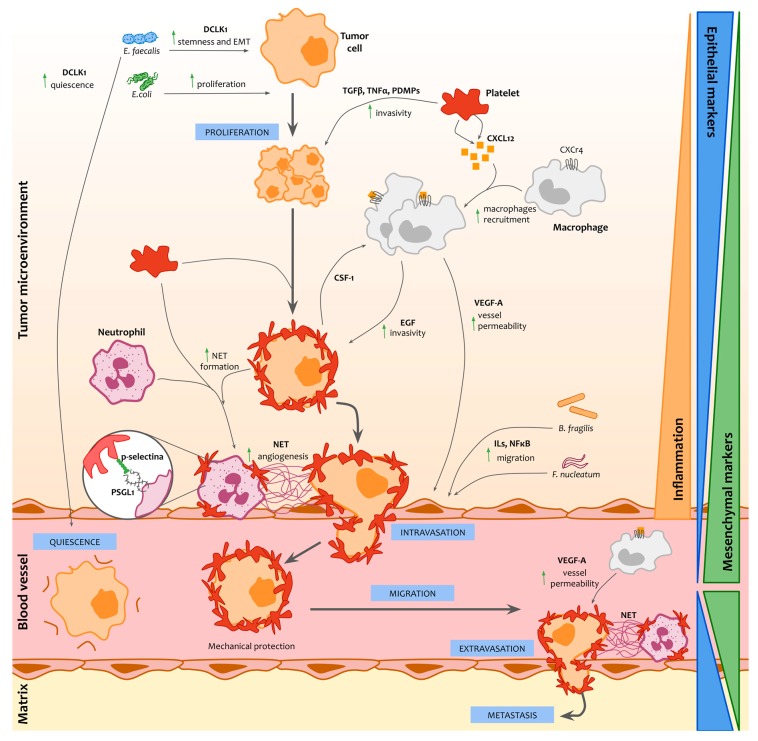
Interactions between circulating tumor cells (CTCs), immune system cells, and microbiome. Metabolites and cytokines produced by bacteria such as *Bacterioides fragilis*, *Enterococcus faecium*, *Escherichia coli*, and *Fusobacterium nucleatum* facilitate proliferation and migration of circulating tumor cells (CTCs), promote stemness and epithelial to mesenchymal transition (EMT), and help CTCs to enter quiescence. Furthermore, platelets interact with proliferating tumor cells directly, by formation of CTCs-platelet complexes allowing CTCs to escape the immune system but also indirectly, through three different ways: secretion of growth factors such as TFGβ, TNFα either alone or enclosed in platelets-derived microparticles (PDMPs) that increase invasivity of CTCs; secretion of chemokines such as CXCL12, increasing macrophages recruitment, what ultimately impact on invasivity and vessel permeability through epidermal growth factor (EGF) and VEFG-A, respectively; and formation of platelet-neutrophil complexes (through P-selectin and PSGL1) that eventually generate neutrophil extracellular traps (NET) promoting angiogenesis and facilitating CTC intravasation to blood vessels. Finally, macrophages and NET also facilitate CTC extravasation from blood vessels to the extracellular matrix to produce metastasis.

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
