# Peer review of "Cooperative and Escaping Mechanisms between Circulating Tumor Cells and Blood Constituents"

_cells, 2019, doi:10.3390/cells8111382_

Round 1
Reviewer 1 Report
The review by Garrido-Navas and colleagues addresses the interaction between circulating tumor cells (CTCs) and the blood constituents by considering immune cells on one side, and the microbiota on the other side.
The interaction of specific monocyte and lymphocyte subpopulations with CTCs has been already addressed by others. At the moment however, this represents a very hot topic as it could provide new strategies to interfere with metastatic dissemination. Unfortunately, Cells just published a review on the same topic two months ago (Never travel alone: the crosstalk of circulating tumor cells and the blood microenvironment).
The idea of exploring data on the role of the microbiota in relation to CTCs is instead new and very intriguing. However, here the authors review literature data that are not directly relevant to the topic. Although the reported do data supports the need of studying CTC also in relation to the patient’s microbiota, they do not answer the question, which the authors their selves are asking asking : ‘How do microbes influence on the inflammation process and on migration and survival of CTCs?’(line 168). The question is extremely interesting, but no answer is provided. Instead a series of literature evidences that support a possible interaction between microbiota and cancer cells, immune cells are listed.
introduction: well written, but I think that reconsidering the references would improve the text. For instance lanes 36-37: maybe it would be better to cite larger studies with greater impact, or mabe even some review promotion of circulating tumor cells through the immune system: this section could be improved by a better organization of the different topics. Consider separate subchapters for neutrophils, platelets etc.. survival of CTCs though the interaction of microbiota with the immune system: no data at all are given demonstrating a direct link between the microbiota-mediated immune cells and CTCs. Maybe it would be better to organize the chapter as a sort of speculation on possible future research avenues, backing it with the cited literature data.
Author Response
The interaction of specific monocyte and lymphocyte subpopulations with CTCs has been already addressed by others. At the moment however, this represents a very hot topic as it could provide new strategies to interfere with metastatic dissemination. Unfortunately, Cells just published a review on the same topic two months ago (Never travel alone: the crosstalk of circulating tumor cells and the blood microenvironment)
The comment of the reviewer is absolutely right. In November 2018 we were invited to collaborate in this issue, we accepted the invitation and then we sent a short abstract with the specific topic that were of our interest for this review (05/04/2019). CELLS accepted the abstract because in their own words “the topic is very interesting and suitable for the special issue.”. We are not sure why two similar papers have been accepted by Cells.
The idea of exploring data on the role of the microbiota in relation to CTCs is instead new and very intriguing. However, here the authors review literature data that are not directly relevant to the topic. Although the reported do data supports the need of studying CTC also in relation to the patient’s microbiota, they do not answer the question, which the authors their selves are asking asking : ‘How do microbes influence on the inflammation process and on migration and survival of CTCs?’(line 168). The question is extremely interesting, but no answer is provided. Instead a series of literature evidences that support a possible interaction between microbiota and cancer cells, immune cells are listed.
In fact, as the reviewer mentions, the potential interaction between the microbiome and CTCs is a very interesting topic although studies up to date are not directly correlated with CTCs. Nevertheless, we explained in different paragraphs the role of the microbiome in the migration, survival and invasion processes of tumour cells according to the literature. For example, in line 180 we explain how E. coli can initiate colon cancer cell invasion and how an unbalanced microbiome can induce activation of the EMT process “an essential step for CTC migration and survival” (lines 200-202 and figure 1). The comment of reviewer is correct regarding absence of studies that correlate microbiome and CTCs. However, in our review, we analyse the bibliography investigating how the microbiome induces migration, invasion and colonization of tumor cells. All these mechanisms necessarily involve CTCs and provide a hypothetical model of interaction between microbiota and CTCs.
introduction: well written, but I think that reconsidering the references would improve the text. For instance lanes 36-37: maybe it would be better to cite larger studies with greater impact, or mabe even some review promotion of circulating tumor cells through the immune system: this section could be improved by a better organization of the different topics. Consider separate subchapters for neutrophils, platelets etc.. survival of CTCs though the interaction of microbiota with the immune system: no data at all are given demonstrating a direct link between the microbiota-mediated immune cells and CTCs. Maybe it would be better to organize the chapter as a sort of speculation on possible future research avenues, backing it with the cited literature data.
According with the recommendation of the reviewer we included and improved the references. However, regarding organization of the topics, we think that separating the information in subchapters will only difficult the reader's understanding of the two main interactions that we want to highlight in this review: between CTCs and immune system and between CTCs and microbiome.
Reviewer 2 Report
The authors present a detailed and timely review of cancer cell evasion through phenotypic change. The review is thorough and well written. I recommend accepting as is.
Author Response
We acknowledge the reviewer's comment
Reviewer 3 Report
The review is very short and deals with the interactions between CTC and other cells in the circulation.
Among the references, some articles are not recently published and are not in well known journals, thus an update of the references is needed.
The review is very compact and short and most parts require to be described more carefully and extensively by the authors.
The introduction on CTCs is limited and does not report the most important papers on the subject.
The authors completely ignore the phenomenon of circulating tumor microemboli that have been studied by several groups, but not cited here.
The interaction of the CTC and microbiota are not sufficiently explained. The microbiota is not circulating in blood as far as I know …
Overall, the action of microbiota on CTCs is not supported sufficiently by the argumentations and references.
Some additional comments:
Line 37 … with reduced disease-free and overall survival. I suggest to cite also some reviews in order to give the authors a general overview of the findings not limited to a specific type of cancer or a single study.
Line 37 ….tumor cells survival is less than 0.01% Please report e reference for this statement.
Line 40-43 It is true that little is known about the dissemination process and the metastatic initiation and in its progression, but some hypothesis and paper have been published already (for example see volume 19 number 11 november 2013 NATURE MEDICINE). Please cite them.
Line47 and 54. Check for repeated words.
Line 51. CTCs and microbes co-evolve inside the ecosystem within our bodies. This sentence requires further explanation in the text. CTCs are in the blood and microbes should be located in different compartments …. Please explain the basis of the co-evolution or reefer to the following paragraphs.
Line 68. Ï•2–4 μM diameter? , cell density? μM microMolar???
Line 69. 1.5–4.0 × 109/L check
Author Response
The review is very short and deals with the interactions between CTC and other cells in the circulation.
The main objective of this review was to present a general overwiew of the principal interactions between immune system, microbiome and CTCs. With this in mind, we wanted to be concise when explaining the relationship between CTCs and other cells analysed by different works without being repetitive or including information previously analyzed in other chapters of this special issue. Obviously, not all biological information was included but we think that we highlighted the most interesting ones, which was accompanied by the specific bibliography.
Among the references, some articles are not recently published and are not in well known journals, thus an update of the references is needed.
According to recommendations of the reviewer we included and update of the references
The introduction on CTCs is limited and does not report the most important papers on the subject.
The main objective of this review was not to give a detailed description on the implications that CTCs have on the development of metastasis because is a topic well addressed previously. However, we intended to highlight the communication process between CTCs and both, immune system and microbiome potentiating the metastatic ability of CTCs.
The authors completely ignore the phenomenon of circulating tumor microemboli that have been studied by several groups, but not cited here.
According to recommendations of the reviewer we have included a sentence and two references regarding tumor microemboli: "In fact, formation of CTC clusters or microemboli, not only composed of CTCs but also leukocytes, cancer-associated fibroblasts, endothelial cells, and platelets, was shown to facilitate the metastatic process and thus be related to poorer outcome in patients with breast [9] and gastric cancer [10] among others."
The interaction of the CTC and microbiota are not sufficiently explained. The microbiota is not circulating in blood as far as I know …
Despite the previously reported fact that the microbiome can circulate in blood (Circulating microbiome in blood of different circulatory compartment Gut 68(3):gutjnl-2018-316227ts.), our objective was not to analyse the circulating microbiome. We tried to analyse the implications of the microbiome in the dissemination, survival and invasion processes of tumor cells. For example, in line 180 we explained how E. coli can initiate colon cancer cell invasion and how the unbalanced microbiome can induce activation of the EMT process “an essential step for CTC migration and survival” (lines 200-202 and figure 1). The comment of reviewer is correct regarding absence of studies that correlate microbiome and CTCs. However, in our review, we analyse the bibliography associated to the different studies, which investigate how the microbiome induces migration, invasion and colonization of tumor cells. All these mechanisms necessarily involve CTCs and provide a hypothetical model of interaction between microbiota and CTCs.
Overall, the action of microbiota on CTCs is not supported sufficiently by the argumentations and references.
The absence of papers on the direct interaction between CTCs and the microbiota emphasizes the difficulty of supporting this interaction. However, we reviewed some aspects in which microbes might impact on CTC survival, migration, transformation and eventually dissemination, suggesting the impact that the microbiome has on CTCs and ultimately in metastasis formation.
Some additional comments:
Line 37 … with reduced disease-free and overall survival. I suggest to cite also some reviews in order to give the authors a general overview of the findings not limited to a specific type of cancer or a single study.
According to recommendations of the reviewer we included reference 3 as review of the role of CTCs in the development of metastasis and thus as indicators of poor prognosis.
Line 37 ….tumor cells survival is less than 0.01% Please report e reference for this statement.
According to recommendations of the reviewer we included this reference.
Line 40-43 It is true that little is known about the dissemination process and the metastatic initiation and in its progression, but some hypothesis and paper have been published already (for example see volume 19 number 11 november 2013 NATURE MEDICINE). Please cite them.
According to recommendations of the reviewer additional and updated references have been included.
Line47 and 54. Check for repeated words.
We have changed "friendly alliance" by "private alliance" in case the word "friendly" seems repetitive with "intimate friendship".
Line 51. CTCs and microbes co-evolve inside the ecosystem within our bodies. This sentence requires further explanation in the text. CTCs are in the blood and microbes should be located in different compartments …. Please explain the basis of the co-evolution or reefer to the following paragraphs.
The sentence on line 51 (now 56) only introduces the topic that is further discussed on section 3 and two references for reviews on the topic are included; however, we have referred to the following paragraphs for a more detailed explanation.
Line 68. Ï•2–4 μM diameter? , cell density? μM microMolar???
In line 68 we used the symbol "Ï•" to refer to platelets size, but to avoid confusion, we changed the symbol "Ï•" by the word "diameter" and we changed the units from "μM" to "μm" to refer to micrometers.
Line 69. 1.5–4.0 × 109/L check
We have re-formatted 1.5–4.0 × 109/L to 1.5–4.0 × 109/L (with superscript) as it was a formatting problem
Round 2
Reviewer 1 Report
I would have prefered to see some effort to try to better clarify the possible interaction of CTCs the microbiota
Author Response
We made our best efforts to review the main pathways in which the microbiome might interact with CTCs. In this review we presented an hypothetical model of interaction, however it was based on an exhaustive review of the main molecular pathways in which the microbiome alters CTC survival and disseminationReviewer 3 Report
Comments for Authors:
The main concern about the submitted review is just on the concept of review itself. In fact, the aim of a review is to report the main results published in the literature on a specific subject. In the rebuttal letter, the authors themselves refers to the absence of papers on the direct interaction between CTCs and the microbiota indicating that their manuscript is not based on data published on this topic by other groups or by themselves.
If this article is a speculation of the authors who believes that "microbes might impact on CTC survival, migration, transformation and eventually dissemination, suggesting the impact that the microbiome has on CTCs and ultimately in metastasis formation", they shall support their hypothesis with data and the present work could be considered as an original article.
For this reason, even though some of the comments have been positively addressed (i.e. a sentence on microemboli), the revised version of the paper does not overcome the lack of supporting evidences for the authors' hypothesis related, in particular, to the hypothetical model of interaction between microbiota and CTCs.
Author Response
Even though there is a lack of studies analysing the direct interaction between CTCs and the microbiome we reviewed sufficient evidences on the impact of the microbiome in pathways directly related with inflammation, EMT process thus having an impact on CTC survival and disemination. We recognise the concern of the reviewer about the nature of this review as we presented an hypothetical model of interaction, however it was based on a exhaustive review of the main molecular pathways in which the microbiome alters CTC survival and dissemination.